# Investigation of Meat from Ostriches Raised and Slaughtered in Bavaria, Germany: Microbiological Quality and Antimicrobial Resistance

**DOI:** 10.3390/biology11070985

**Published:** 2022-06-29

**Authors:** Philipp-Michael Beindorf, Oksana Kovalenko, Sebastian Ulrich, Hanna Geißler, Rüdiger Korbel, Karin Schwaiger, Samart Dorn-In, Irene Esteban-Cuesta

**Affiliations:** 1Chair of Food Safety and Analytics, Veterinary Faculty, LMU Munich, 85764 Oberschleissheim, Germany; philipp.beindorf@ls.vetmed.uni-muenchen.de (P.-M.B.); medvet.kovalenko@gmail.com (O.K.); hanna.geissler@ls.vetmed.uni-muenchen.de (H.G.); 2Chair of Bacteriology and Mycology, Institute for Infectious Diseases and Zoonosis, Department of Veterinary Sciences, Faculty of Veterinary Medicine, LMU Munich, 80539 Munich, Germany; ulrich@micro.vetmed.uni-muenchen.de; 3Centre for Clinical Veterinary Medicine, Clinic for Birds, Small Mammals, Reptiles and Ornamental Fish, LMU Munich, 85764 Oberschleissheim, Germany; korbel@vogelklinik.vetmed.uni-muenchen.de; 4Unit of Food Hygiene and Technology, Institute of Food Safety, Food Technology and Veterinary Public Health, University of Veterinary Medicine Vienna, 1210 Vienna, Austria; karin.schwaiger@vetmeduni.ac.at (K.S.); samart.dorn-in@vetmeduni.ac.at (S.D.-I.)

**Keywords:** antimicrobial resistance, meat microbiology, *Salmonella*, STEC, *Trichinella*

## Abstract

**Simple Summary:**

The rapid increase in the world population might lead to the need for new food resources. Ostrich meat remains an exotic food product, although it is characterized by high nutritional value. However, to secure food safety, good monitoring and control programs have to be implemented. Furthermore, standard data concerning the general microbiological status of meat, its evolution in time, and aspects influencing the different parameters have to be assessed in order to provide current information. Thus, production, slaughter, and processing procedures may be improved. For this purpose, the microbiological status, prevalence of zoonotic pathogens, and presence of antimicrobial-resistant microorganisms were assessed. The results will provide baseline data that can be used for future official specific hygiene control procedures and monitoring programs for producing ostrich meat.

**Abstract:**

Ostrich meat is characterized by high nutritional value; however, it remains an exotic product in most countries worldwide. In Europe, only few data are available regarding its microbial contamination, prevalence of antimicrobial-resistant bacteria, and safety. Therefore, this study aimed to investigate the microbiological quality and safety of ostrich meat samples (*n* = 55), each from one animal, produced in Bavaria, Germany. The provided microbiological status of ostrich meat included mesophilic aerobic bacteria, Enterobacteria, and mesophilic yeast and molds. In terms of food safety, all meat samples were negative for *Salmonella* spp. and *Trichinella* spp. Additionally, meat samples and a further 30 stool samples from 30 individuals were investigated for Shiga toxin-producing *Escherichia coli* genes, with two meat samples that were qPCR-positive. Antimicrobial-resistant *Enterobacteriaceae*, *Enterococcus faecalis*, and *Enterococcus faecium* strains were from meat and stool samples also analyzed; 13 potentially resistant *Enterobacteriaceae* (meat samples) and 4 *Enterococcus faecium* (stool samples) were isolated, and their susceptibility against 29 and 14 antimicrobials, respectively, was characterized. The results of this study provide an overview of microbial loads and food safety aspects that may be used as baseline data for the ostrich meat industry to improve their hygienic quality. However, the implementation of monitoring programs is recommended, and microbiological standards for ostrich meat production should be established.

## 1. Introduction

Ostrich meat is recognized as a product with high nutritive value. It can be a healthy alternative to other red meats since it contains a highly favourable fatty acid profile but a low intramuscular fat content [1,2]. In comparison to other continents, ostrich farming in Europe is a relatively young sector, with Poland being one of the main producers, followed by Italy and Spain [3,4]. In Germany, about 2500 ostriches are registered, and according to the German State Ministry for Environment and Consumer Protection’s report, 48 ostrich farms with a total of 2103 birds were registered in Bavaria in 2018 [4,5]. Almost all ostrich farms in Germany have their own abattoirs, which are usually located close to the animal raising area. The ostrich farms and abattoirs are generally approved by German local authorities, in order to ensure that raising and slaughtering processes conform with animal welfare legislation and general hygienic standards. Ostrich meat is generally sold fresh and, in some cases, deep-frozen. Although this meat may be cooked for consumption, there are many products such as carpaccio that are prepared with raw meat and might be a source of infection.

In terms of microbiological quality of ostrich meat and products, as well as related to the topic of antimicrobial-resistant (AMR) microorganisms, only few data are available [6,7,8,9]. Therefore, the aim of this study was to determine the microbiological quality of ostrich meat after slaughter and before any further processing and to provide a status quo for this animal and product regarding microbiological quality, including certain food safety parameters such as Shiga toxin-producing *Escherichia coli, Trichinella* spp., and *Salmonella* spp., and prevalence of antimicrobial-resistant bacteria. The resulting data shall serve as a reference in future investigations.

## 2. Materials and Methods

### 2.1. Sample Collection

A total of 55 meat and 30 stool samples originating each from one individual were collected from four ostrich farms in Bavaria (Germany). Meat samples (approximately 50 g) were taken immediately after slaughter and skinning from the ventral part of the *musculus gastrocnemius*, since this is the preferential distribution site of *Trichinella* spp. larvae [10] and was easily accessible for sampling. This research did not involve live animals. Stool samples were collected from the colon *post mortem* and directly transferred in 50 mL falcon tubes. All samples were transported under refrigerated conditions to the laboratory within six hours. Directly upon arrival, meat samples were investigated for the presence of *Trichinella* spp. The remaining meat and stool samples were frozen at −20 °C until they were investigated for their microbiological quality and prevalence of antimicrobial-resistant bacteria.

### 2.2. Detection of Trichinella spp.

Meat samples were analyzed for the presence of *Trichinella* spp. using artificial digestion and serological analysis. The digestion method described in the European Regulation (EU) 2015/1375 was adapted for the analysis of ostrich meat, and five grams of the *musculus gastrocnemius* was used. As a positive control, a *Trichinella*-positive pork sample was provided by the German Federal Institute for Risk Assessment (BfR, Berlin, Germany). 

For serological analysis, another five grams of ostrich meat was used for each sample, and the commercially available ELISA BioRad Trichin-L Ag Test Kit (Bio-Rad Laboratories GmbH, Feldkirchen, Germany) was applied. Sample preparation, analysis, and interpretation of the results were carried out according to the manufacturers’ instructions.

### 2.3. General Microbiological Status

The preparation of meat and stool samples was carried out in accordance with the methods described by the International Organization of Standardization (ISO). Ten grams of each sample were homogenized with 90 mL buffered peptone water (BPW; Merck KGaA, Darmstadt, Germany) using a stomacher. The homogenized suspensions were subsequently used for all microbiological analyses, namely quantitative microbiological analysis of the total mesophilic aerobic bacteria (MAB; ISO 4833-2:2013-12), *Enterobacteriaceae* (ISO 21528-2:2004; anaerobic incubation: O_2_ < 0.1%, CO_2_ 7.0–15.0%), and mesophilic yeasts and molds (ISO 10186:2005-10; modified for food using the spatula method). For *Salmonella* spp., a qualitative microbiological analysis according to the ISO Norm 6579-1:2017 was applied. 

### 2.4. Detection of the Shiga Toxin-Producing Escherichia coli Genes stx_1_ and stx_2_ Including stx_2f_

The STEC genes *stx_1_*, *stx_2_,* and *stx_2f_* were investigated using an in-house multiplex qPCR. Table 1 shows sequences and references of specific primer pairs and probes for each target gene. The qPCR probe for the *stx_2f_* gene was newly designed in this study, since this gene was found in avian species (pigeons) and could therefore be of interest in ostriches [11]. For this, a target sequence available at GenBank (NCBI, accession No. AJ010730.1) was used and combined with the primer pair developed by Scheutz et al. [12]. Additionally, both the primer and the probe for internal amplification control (IAC) were included in each PCR reaction.

An enrichment protocol was applied in order to obtain a sufficient amount of STEC for further analyses. Meat samples were mixed 1:10 with modified Tryptic Soy Broth (mTSB, Merck KGaA, Darmstadt, Germany) and stool samples with BPW. The suspensions were incubated in a water bath at 37 °C with shaking at 99 rpm for 18 h. The enrichment was subjected to DNA extraction using a commercially available ISOLATE II Genomic DNA Kit (Bioline GmbH, Luckenwalde, Germany). One milliliter matrix suspension was filled in a 1.5 mL reaction tube and centrifuged at 8000× *g* for 5 min. The supernatant was discarded, and the pellet was resuspended with 180 µL lysis buffer and 25 µL proteinase K (provided by DNA extraction kit) and incubated at 56 °C for 1 h in a thermomixer (Eppendorf SE, Hamburg, Germany). The subsequent steps were performed according to the instructions of the DNA extraction kit. 

The reference STEC strains C600J1, C600W34, and MHI 832 were used as positive controls and qPCR calibration standards for the genes *stx_1_*, *stx_2_,* and *stx_2f_*, respectively. The strains were subcultured on blood agar (OXOID Deutschland GmbH, Wesel, Germany) and incubated at 37 °C for 24 h. For each strain, one colony was transferred into 4 mL of 0.9% sterile sodium chloride (NaCl) using a sterile inoculation loop. The cell suspension was adjusted to a 1.0 McFarland turbidity standard (approximately 3.0 × 10^8^ CFU g^−1^) [17,18]. Then, 0.1 mL cell suspension was serially diluted in 0.9% NaCl up to the dilution 10^−6^. The original McFarland suspension was stored at 4 °C to prevent the growth of STEC strains, while 100 µL from dilutions 10^−4^ to 10^−6^ were plated on Casein–Soja–Pepton agar (Merck KGaA, Darmstadt, Germany) and incubated at 37 °C for 24 h under aerobic condition, and then plate counting was performed. After that, the concentration of each STEC strain in each original suspension was adjusted to 10^7^ CFU mL^−1^ in NaCl. Then, all three STEC suspensions were equally mixed, resulting in a concentration of 3.3 × 10^6^ CFU mL^−1^ for each STEC strain. This bacterial solution was inoculated on each matrix suspension (meat and stool) to correspond to 3.3 × 10^6^ CFU g^−1^ of sample. Subsequently, all matrix suspensions (with and without artificial contamination with the STEC strains) were subjected to DNA extraction using the ISOLATE II Genomic DNA Kit (Bioline GmbH, Luckenwalde, Germany). 

For quantification of the target genes in the meat and stool samples via qPCR, a 10-fold dilution of the extracted DNA was prepared. The concentration of the bacterial strains for each target gene in the standard dilution ranged from 3.3 × 10^6^ to 3.3 × 10^2^ CFU g^−1^ meat or stool. Each reaction of a multiplex qPCR contained a 20 μL mixture that included 10 μL SensiFAST Probe No-ROX (Bioline GmbH, Luckenwalde, Germany), 0.2 μM of each primer, 0.07 µM of each probe, 2 µL DNA sample, and 1 µL pUC19 (100 copies for IAC; Affymetrix, Thermo Fisher Scientific, USA) and was filled up with molecular grade water. The qPCR run was performed in Bio-Rad CFX96 Touch (Bio-Rad Laboratories GmbH, Feldkirchen, Germany), starting with an initial denaturation of 5 min at 95 °C and 45 cycles of denaturation at 95 °C for 5 s, annealing, and elongation at 60 °C for 30 s.

### 2.5. Phenotypical Antimicrobial Susceptibility of Enterobacteriaceae and Enterococcus spp.

Isolates intended for the antimicrobial susceptibility analysis were *Ent. faecium* or *Ent. faecalis* strains grown on Slanetz–Bartley (SB) agar according to DIN 10106:2017-04 and potentially resistant *Enterobacteriaceae* (grown on MacConckey No. 3 Agar containing 1 μg/mL cefotaxime sodium salt (Mac+)). Both meat (*n* = 55) and stool (*n* = 30) samples were investigated. Microbial species identification was performed using matrix-assisted laser desorption/ionization time-of-flight mass spectrometry (MALDI-TOF MS). All measuring parameters were set according to the Bruker Daltonik GmbH Guidelines [19] and as described by Esteban-Cuesta et al. [20]. Identification of the protein spectra was performed using the Biotyper OC-Software 3.0 reference spectra database (Bruker Daltonik GmbH, Bremen, Germany). 

Microorganisms grown on SB agar and identified as *Ent. faecalis* or *Ent. faecium* were analyzed for their antimicrobial susceptibility using Sensititre Gram Positive AST and Sensititre Companion Animal Gram Positive Vet AST Plates (Thermo Fischer Scientific GmbH, Dreieich, Germany), which included a total of 29 antimicrobials (Appendix A, Table A1). For this analysis, *Ent. faecalis* ATCC 29212 and *Pseudomonas aeruginosa* ATCC 27853 were used as positive and negative controls, respectively

To examine the potentially resistant *Enterobacteriaceae* grown on Mac+ agar, Brilliance Extended-Spectrum Beta-Lactamase (ESBL) agar (OXOID Deutschland GmbH, Wesel, Germany) and Minimum Inhibitory Concentration (MIC) Test Strips ESBL (Liofilchem S.r.l., Roseto degli Abruzzi TE, Italy) were used to assess ESBL resistance of the suspicious strains (Appendix A, Table A2). Furthermore, Sensititre NARMS Gram-negative Plates (Thermo Fischer Scientific GmbH, Dreieich, USA) were used to test sensitivity to a total of 14 antimicrobials (Appendix A, Table A3). As positive and negative controls, *Klebsiella pneumoniae* ATCC 700603 and *E. coli* ATCC 25922, respectively, were used.

## 3. Results and Discussion

### 3.1. Detection of Trichinella spp.

*Trichinella* spp. infestations in humans are observed all over the globe in most climate circumstances, and *Trichinella* spp. are among the most widespread parasites [21]. Trichinosis in humans usually occurs after ingestion of raw or undercooked meat infected with larvae of this parasite, even at a very low concentration such as one larva per gram of meat for *Trichinella spiralis* [22]. Since an experiment from Fioretti et al. [10] showed that ostriches are susceptible to *Trichinella pseudospiralis* when they were artificially infected with this parasite, its naturally occurring prevalence in ostrich meat was investigated in this study. 

To confirm the results obtained by the standard artificial digestion method, a serological method was additionally applied. All tested samples were negative for *Trichinella* spp. by means of both detection methods, while the positive control showed a correspondingly positive value. Nevertheless, the German Federal Institute for Risk Assessment (BfR) concluded that there is no region in Germany that may be considered as of “negligible risk” for *Trichinella* spp. [23]. Additionally, since farm ostriches are kept under extensive conditions and raised free-range and their production protocols and pathways are much less standardized than those for commercial kept poultry (i.e., broiler chicken industry), they may be infected via various vectors such as wild animals. The risk of infection with *Trichinella* spp. has to be considered and monitored according to their possible contacts with other carriers of *Trichinella* spp.

### 3.2. Microbiological Examination

A total of 42 meat samples (76.4%) were found positive for mesophilic aerobic bacteria (MAB, Figure 1). Colony counts of the positive meat samples ranged from 2.0 to 5.5 log CFU g^−1^. MAB are used as a hygienic parameter of the slaughtering process (European Commission Regulation (EC) No. 2073/2005). Since ostrich farming and slaughtering in the European Union (EU) is relatively new and rare, explicit microbiological criteria for ostrich meat have not been established yet. Therefore, the existing MAB criterion value for carcasses of cattle, sheep, goats, and horses (5.0 log CFU cm^−^²) was chosen for ostrich meat, as these animals are also skinned after slaughter. In this study, whole muscle pieces were taken for the investigation, as the number of bacteria recovered through swabbing strongly varies depending on different factors, and usually, only loosely attached bacteria are sampled [24]. However, the contamination level per square centimetre in cored meat samples would be expected to be lower than on the surface of carcasses (per square centimetre), since the internal part of muscle tissue should remain free of contamination. Although a direct comparison between CFU cm^−2^ and CFU g^−1^ is not directly possible, we will try to present them according to the EU regulation, as this is the only official guideline available. Therefore, following the MAB values in this EU regulation, MAB contamination in two samples (4.8%) exceeded this given criterion. In some cases, the MAB contamination level found in this study is similar to the contamination level found on post-eviscerated ostrich carcasses reported in South Africa (4.21 ± 0.63 CFU cm^−^²) [25] or in meat already handled and sampled prior to storage and investigated in Slovakia (3.1 log CFU g^−1^, [26]) and Poland (3.8 ± 0.3 log CFU g^−1^, [27]). 

Similar to MAB*,*
*Enterobacteriaceae* are used as a process hygiene parameter according to EU Regulation (EC) No. 2073/2005. Additionally, they were considered as a food safety indicator, since some species belonging to this family are pathogenic [28]. In this study, *Enterobacteriaceae*-positive meat samples (*n* = 6; 10.9%) showed a contamination level ranging from 1.8 to 3.4 log CFU g^−1^. Other studies in ostrich meat show variable results, ranging from not detected [6,26] and a very low amount on the carcass surface (1.8 and 1.5 log CFU/2500 cm^2^ for coliforms and *E. coli,* respectively [29], to up to 91% of the carcasses being contaminated with *E. coli* [9], or even all tested ostrich meat samples being positive for *Enterobacteriaceae* [25]. The level of contamination with *Enterobacteriaceae* is a measure of the hygiene level that may vary between ostrich abattoirs, especially during the skinning and evisceration process. According to the EU Regulation (EC) No. 2073/2005, the critical value of contamination of *Enterobacteriaceae* set for carcasses of cattle, sheep, goats, and horses is 2.5 log CFU cm^−2^ (daily mean value). In this study, a high level of *Enterobacteriaceae* (3.38 log CFU g^−1^) was found in a single meat sample, suggesting even higher levels on the carcass surface. 

Mesophilic yeasts were isolated from 32 meat samples (58.2%) with a load ranging from 1.3 to 4.2 log CFU g^−1^. The scope for mesophilic molds in positive meat samples (38.2%) ranged from 1.3 to 3.3 log CFU g^−1^. Altogether, fungi were found in a relatively high concentration in this study, while Hoffman et al. [6] could isolate only a few yeasts in ostrich meat in the post-chilling phase. Since criteria for fungal contamination in fresh meat are not yet established by the European Union, the evaluation of meat quality concerning this topic was not performed. 

*Salmonella* spp. are amongst the most relevant foodborne pathogens in the European Union (EU) and worldwide. Foodborne outbreaks involving *Salmonella enterica* subsp. *enterica* in the EU are regularly associated with poultry [30,31]. As an avian species, ostriches are susceptible to *Salmonella* spp. infection: results from other studies show that 18.5% of living ostriches were infected with *Salmonella* spp. [32], while a high variation in the prevalence in different parts of the intestinal tract and feces (gizzards 5%, skins 8.3%, large intestines 26.2%, small intestines 16.1%, and feces 44.2%) was observed [7]. Another study could only detect one *Salmonella-*positive carcass sample (0.65%) after handling in slaughterhouses in the US [9]. In the current study, all meat and stool samples were negative for *Salmonella* spp., corresponding to the food safety criterion of EU Regulation (EC) No. 2073/2005, which states that meat must be free from *Salmonella* spp.

### 3.3. Detection of the Shiga Toxin-Producing E. coli Genes stx_1_ and stx_2_ including stx_2f_

The presence of STEC genes in ostrich meat poses a health risk for the consumer if the meat is not sufficiently heated before consumption. Infection with Shiga toxin-producing *E. coli* (STEC) was the third most reported zoonosis in humans in 2019, and it increased in prevalence from 2015 to 2019 [31]. The most common source of infection was the meat of different types derived from different animal species, but especially cattle [31]. Besides direct infection, STEC can additionally transfer the *stx* genes to other non-STEC *E. coli* via horizontal gene transfer through bacteriophages [33].

All stool samples were negative for the analyzed STEC genes. Two out of fifty-five meat samples (3.64%) tested positive, one for the gene *stx_1_* (approximately 7.5 log CFU g^−1^ enrichment) and the other one for both *stx_1_* and *stx_2_* (approximately 2.0 and 3.2 log CFU g^−1^ enrichment, respectively). The limit of detection using the established multiplex qPCR protocol was 3.3 × 10^2^ CFU g^−1^ for genes *stx_1_* and 3.3 × 10^3^ CFU g^−1^ for genes *stx_2_* and *stx_2f_.* However, STEC strains could not be isolated from any positive meat samples. The isolation of STEC is considered challenging, and the detection of the genes is a common phenomenon but the successful culture of the isolates is lacking [34]. Consequently, diagnosis and detection of STEC by means of qPCR are increasingly being considered sufficient [31]. 

### 3.4. Phenotypical Antimicrobial Susceptibility

*Enterococcus* spp. possess a strong tendency to acquire antimicrobial resistance; thus, they are considered important key indicator bacteria and have been included in several human and veterinary resistance surveillance systems [35,36,37,38,39]. In this study, *Enterococcus* spp. were detected in 5.5% (*n* = 3/55) of the meat samples with a concentration of 1.30 log CFU g^−1^ in all three samples. In contrast, 90% of the stool samples were found positive with a load ranging from 1.60 to 6.67 log CFU g^−1^, and the mean value was 3.35 log CFU g^−1^. Although there are no data available on this for ostriches, another study on cattle could find 53.9% positive stool samples for *Enterococcus* spp., although high variation between sampling days was observed, ranging from 30 to 95% positive samples [40]. In contrast, Fluckey et al. [40] found 58.3% positive cattle carcass samples.

Only isolates identified as *Ent. faecium* or *Ent. faecalis* (if present) were further analyzed for their phenotypical antimicrobial susceptibility, since these species are listed in the surveillance program of the EARS-Net [41]. By means of MALDI-TOF MS, *Ent. faecium* was identified in 4 of 30 (13.3%) stool samples, while *Ent. faecalis* was found in neither meat nor stool samples. Although no information is available on ostriches, Bortolaia et al. [42] reviewed the prevalence of *Ent. faecium* and *Ent. faecalis* in raw poultry meat, finding a prevalence that ranged between 0 and 96%, showing a strong variability among countries worldwide. 

All four *Ent. faecium* strains were tested against 29 antimicrobials, and the minimal inhibitory concentration (MIC) values are shown in Appendix A, Table A1. Epidemiologic cut-off (ECOFF) values are available for 15 of the tested antimicrobials but were only reached against daptomycin (*n* = 4; ECOFF = 8 µg/mL), imipenem (*n* = 1; ECOFF = 4 µg/mL), and levofloxacin (*n* = 2; ECOFF = 4 µg/mL) and exceeded against erythromycin (*n* = 4; ECOFF = 4 µg/mL). Although all four isolates were only inhibited with the highest nitrofurantoin concentration tested (64 µg/mL), the ECOFF value is set at 256 µg/mL for *Ent. faecium* for this antimicrobial, and therefore they were considered not resistant. The tentative (T) ECOFF value for the fluoroquinolone moxifloxacin (1 µg/mL) was exceeded by two isolates. Although no ECOFFs are set for the remaining antimicrobials, at least one strain showed no susceptibility against the highest antimicrobial concentrations tested for amikacin (*n* = 1; MIC >32 µg/mL), cefazolin (*n* = 4; MIC >4 µg/mL), cefovecin (*n* = 4; MIC >8 µg/mL), cephalothin (*n* = 4; MIC >4 µg/mL), cefpodoxime (*n* = 4; MIC >8 µg/mL), ceftriaxone (*n* = 3; MIC >64 µg/mL), clindamycin (*n* = 2; MIC >4 µg/mL), oxacillin + 2% NaCl (*n* = 4; MIC >2 µg/mL), and rifampicin (*n* = 4; MIC >2 µg/mL). Among these, three third-generation cephalosporins (3GCs) are included, of which cefovecin is only intended for veterinary use. We also tested resistance against the only existing 5GC, ceftaroline, which can be used in combination with daptomycin for the treatment of non-daptomycin-susceptible vancomycin-resistant Enterococci [43]. All samples showed an MIC value of 0.5 µg/mL, which still remains below the ECOFF/TECOFF values set for other bacterial genera (1 µg/mL), as none is set for *Enterococcus* spp. Thus, the strains tested were susceptible to vancomycin. All strains were chloramphenicol- and vancomycin-susceptible (ECOFFs set at 32 µg/mL and 4 µg/mL, respectively). *Enterococcus* spp. strains isolated from cattle showed also resistance against erythromycin and susceptibility against vancomycin and chloramphenicol [40]. *Enterococcus* spp. are intrinsically resistant to many antimicrobials, but their resistance pattern also varies strongly among countries and strains. For example, in poultry, ampicillin resistance may vary from 3% in *Ent. faecium* from domestic broiler meat in Denmark to 10 and 54% of the isolates from chicken and turkey meat in the United States, respectively. Additionally, it seems that also the low prevalence of vancomycin-resistant enterococci was present in raw poultry meat in Europe, but was higher in the United States [42].

Concerning the analyzed *Enterobacteriaceae,* a total of 13 potentially resistant strains were isolated from 12 meat samples (21.8%) by means of Mac+ agar. No stool samples were positive for potentially resistant *Enterobacteriaceae.* From these, 12 isolates grew on Brilliance ESBL agar and were identified by means of MALDI-TOF MS as *Klebsiella aerogenes* (*n* = 2), *Enterobacter cloacae* (*n* = 9*)*, and *Pantoea septica* (*n* = 1). These bacterial strains were further analyzed using MIC Strip Test ESBL (Liofilchem, Italy; see Appendix A, Table A2). All 12 isolates remained non-determinable (ND) according to the manufacturers’ instructions. However, one sample (*Klebsiella aerogenes*) could be considered ESBL-positive for the ceftazidime/ceftazidime with CA strip, although ND for the other antimicrobials; hence, further analysis for confirmation would be necessary. Diminished susceptibility was shown by 11 isolates against cefotaxime and by all 12 isolates against its combinations with CA. Nine isolates showed increased susceptibility against ceftazidime, and ten against its combinations with CA. All isolates were susceptible to cefepime and its CA combination.

The 13 positive isolates on Mac+ agar isolates were also further tested against 14 antimicrobials using SENSITITRE MIC susceptibility plates (see Appendix A, Table A3). Some of the isolates showed low susceptibility against cefoxitin (*n* = 12), amoxicillin/clavulanic acid (*n* = 7), ampicillin (*n* = 12), ceftriaxone (*n* = 6), and azithromycin (*n* = 1). All these antimicrobials are included in the list of critically important antimicrobials by the WHO, with exception of cefoxitin [44]. Nevertheless, cefoxitin resistance can be decisive in the detection of AmpC-resistant enterobacteria. *AmpC* genes could be detected in most *En. cloacae* isolates analyzed in a previous study [20], which were resistant to cefoxitin, amoxicillin/clavulanic acid, and ampicillin. Cefoxitin resistance can be enhanced by the reduction in outer membrane permeability of the isolates [45]. However, further genotypic tests are required for confirmation of *AmpC* genes. Some isolates also showed low susceptibility against further critically important antimicrobials, such as gentamicin (*n* = 2) and nalidixic acid (*n* = 1). Both *K. aerogenes* isolates had MIC values of 2 µg/mL for the antimicrobial gentamicin, which overlaps with the ECOFF value set by the EUCAST. Although no ECOFF value is set for *P. septica* against nalidixic acid, other *Enterobacteriaceae* such as *E. coli* and *Salmonella enterica* have an ECOFF value of 8 µg/mL [44], which was reached in this study. A summary of all the results of the MIC susceptibility test is shown in Appendix A, Table A3.

With regard to antimicrobial use in ostriches in Germany, there are no antimicrobials that are registered for legal use in ostriches. Therefore, to treat ostriches, antimicrobials that are legal for use in turkey, chicken, and waterfowl have to be derived according to the “rededication cascade (Umwidmungskaskade)”, and these can only be applied in case of a treatment emergency [46,47,48]. Nevertheless, according to the long-term experience of the author (RK) in establishing an experts’ seminar in ostriches, rheas, and emus [49], as well as from clinical work with one of the largest ostrich farms (*n* = 800–1000) in Bavaria, antimicrobial treatment was not necessary for systemic or flock-related infections. The only use of antimicrobials in ostriches in Bavaria was, according to the authors’ experience, to treat predator bite lesions (e.g., fox bites), where the maximum waiting times applied. No side effects were observed. Treatment of chicks (5 to 14 days of age) is only considered necessary with benzylpenicillin when treating clostridiosis and only in flocks with a high percentage of losses [49,50].

## 4. Conclusions

Microbiological data (MAP and *Enterobacteriaceae*) of almost all meat samples tested were considered as corresponding to the criteria adopted from the European Union (Regulation (EC) No. 2073/2005), although values in the study are meant per gram and not per square centimeter, as required in the EU regulation. *Salmonella* spp. were not detected in meat or stool samples. *Trichinella* spp. were not detected in any meat sample. In terms of food safety, two meat samples were considered a potential hazard for foodborne infection, since they were positive for STEC genes responsible for the production of Shiga toxins. Additionally, four *Ent. faecium* strains isolated from stool samples reached the ECOFF values set by the EUCAST for some antimicrobials and showed low susceptibility against others without (T)ECOFF values. Furthermore, 13 *Enterobacteriaceae* isolated from meat samples showed resistance and/or diminished susceptibility against antimicrobials considered critically important by the WHO. Results from stool analysis did not necessarily correspond to those from meat analysis, so conclusions concerning the transmission of pathogens or prevalence of antimicrobial-resistant bacteria cannot be extrapolated. The data obtained from this study provides information on the microbiological quality and safety aspects of ostrich meat produced in Bavaria, Germany. A specific microbiological monitoring program in order to improve the hygienic status of ostrich farming and slaughtering should be implemented. For this, more studies concerning the microbial quality of ostrich meat from farms in Germany and Europe should be performed, so that a unified monitoring program is available in the European Union. Additionally, more data on the prevalence of AMR microorganisms in retail ostrich meat compared with ostrich meat after slaughter would also be of main interest, especially considering the strong variations that might be present among countries, and in order to acknowledge the routes of dissemination that are of main importance for this animal and food product. 

## Figures and Tables

**Figure 1 biology-11-00985-f001:**
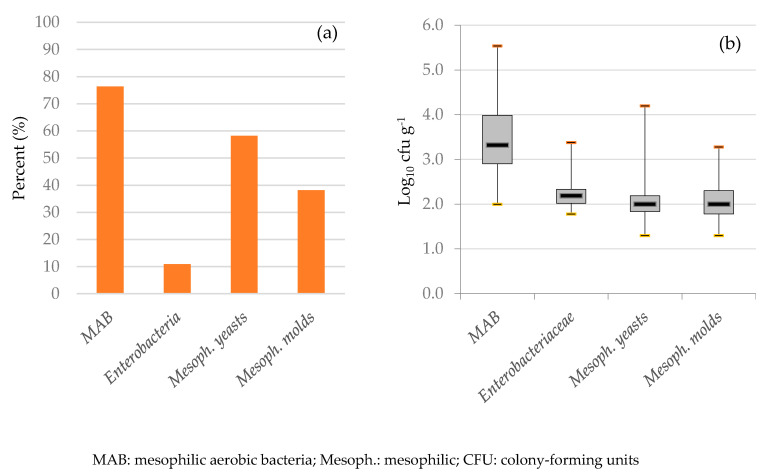
(**a**) Percentage of positive samples for the analyzed parameters in the microbial analysis and (**b**) microbial load of positive samples (line: mean, box: 25th–75th percentile, whisker: box +/− 1.5 interquartile range).

**Table 1 biology-11-00985-t001:** Primers and probes used in the multiplex qPCR to detect the *stx_1_*, *stx_2_*, and *stx_2f_* genes of *Escherichia coli*.

Target Gene	Primer/ Probe	Sequence (5′–3′)	Fragment Size (bp)	Reference
*stx* _1_ */* *stx* _2_	*stx*_1+2_-F	TTTGTYACTGTSACAGCWGAAGCYTTACG	131/128	[13,14]
	*stx*_1+2_-R	CCCCAGTTCARWGTRAGRTCMACRTC		
	*stx*_1_-Probe	FAM-CTGGATGATCTCAGTGGGCGTTCTTATGTAA-BHQ1		
	*stx*_2_-Probe	TEXASRED-TCGTCAGGCACTGTCTGAAACTGCTCC-BHQ2		
*stx* _2*f*_	*stx*_2*f*_-F	TGGGCGTCATTCACTGGTTG	424	Primers: [12,15]
	*stx*_2*f*_-R	TAATGGCCGCCCTGTCTCC		
	*stx*_2*f*_*-*Probe	Cy5.5-ATTCCGACCGGCGCTGTCTGAG-BBQ-650		Probe: this study
IAC *	IAC-F	GCAGCCACTGGTAACAGGAT	118	[13,16]
	IAC-R	GCAGAGCGCAGATACCAAAT		
	IAC-Probe	HEX-AGAGCGAGGTATGTAGGCGG-BHQ1		

* IAC: Internal amplification control.

## Data Availability

Not applicable.

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
