# Peer review of "Investigation of Meat from Ostriches Raised and Slaughtered in Bavaria, Germany: Microbiological Quality and Antimicrobial Resistance"

_biology, 2022, doi:10.3390/biology11070985_

Round 1

Reviewer 1 Report

The article is well written and has relevance. I suggest to authors that they better explore their introduction, in order to justify their research. Present numerical results of the nutritional composition of ostrich meat, which fatty acids are in greater proportion in this meat? The polyunsaturated ones? How is the consumption of this meat by the consumer market today? The ways of selling? What is the relevance of the search? Why carry out this study? Is it innovative? Do you already have research with this object of study? What makes this research different from previously published articles?

What is the research hypothesis?

After the hypothesis, the authors must insert the objective

This entire written part should be removed: "For this purpose, quantitative and qualitative investigation of microorganisms on ostrich meat was performed, namely mesophilic aerobic bacteria, Enterobacteriaceae, Salmonella spp., Shiga-toxin producing Escherichia coli, and mesophilic yeasts and molds. Since Furthermore, meat and stool samples were analyzed for antimicrobial resistant (AMR) Enterobacteriaceae and Enterococcus spp. The results of this study provide a status quo for this animal and product regarding microbiological quality and food safety, which may serve as reference in future investigations."This item is for the introduction/justification of the article. Authors end their introduction as if they were finalizing the presentation of their research project and this is totally wrong.

Author Response

Submission of a revised manuscript biology-1785144

Responses to the reviewer’s comments

Dear Reviewer,

thank you very much for investing you time in this revision and for your comments. Please find the point by point response to your suggestions below.

Reviewer #1:

The article is well written and has relevance. I suggest to authors that they better explore their introduction, in order to justify their research.

Present numerical results of the nutritional composition of ostrich meat, which fatty acids are in greater proportion in this meat? The polyunsaturated ones?

R: Dear reviewer, we did not include this information, because the numbers vary between studies, animal, ostrich breed and muscle analyzed. Therefore, we do cite two publications that exclusively focus on that topic and present numbers to the different breeds and muscles that may be use for meat.

How is the consumption of this meat by the consumer market today? The ways of selling?

R: We include information on how it might be purchased and consumed between L58 and L61

What is the relevance of the search? Why carry out this study? Is it innovative? Do you already have research with this object of study? What makes this research different from previously published articles?

R: We have revised the sentence on L72 to make the aim of the study clear. As stated on the manuscript, there is very few descriptive data available for these animals, and this can only be provided by this kind of studies. There are almost no studies in Europe to this topic and none in Germany, which makes it therefore new and necessary (L62-64). Additionally, no previously studies provide such a complete information in so many different aspects of food microbiology, prevalence of antimicrobial resistance and food safety. It can be noted in how few references can be provided on this topic. Therefore, we think that this information might be of interest in future research studies in ostriches, and ostrich meat and products thereof (L72-74).

What is the research hypothesis? 

R: Thanks for your comment. As this is an exploratory/descriptive study, no hypothesis has been exposed.

After the hypothesis, the authors must insert the objective

R: Sentence on L72-74 have been revised so that the aim of the study is clear.

This entire written part should be removed: "For this purpose, quantitative and qualitative investigation of microorganisms on ostrich meat was performed, namely mesophilic aerobic bacteria, Enterobacteriaceae, Salmonella spp., Shiga-toxin producing Escherichia coli, and mesophilic yeasts and molds. Since Furthermore, meat and stool samples were analyzed for antimicrobial resistant (AMR) Enterobacteriaceae and Enterococcus spp. The results of this study provide a status quo for this animal and product regarding microbiological quality and food safety, which may serve as reference in future investigations."This item is for the introduction/justification of the article. Authors end their introduction as if they were finalizing the presentation of their research project and this is totally wrong.

R: The sentence was erased, as requested.

Reviewer 2 Report

The manuscript aimed to investigate the microbiological quality and safety of ostrich meat samples. The study provides very useful information for the microbiocidal quality and safety of ostrich meat. However, the data provided in the manuscript was not sufficient, only 50 meat samples were used, and only 3 safety indicators were tested, other important pathogens such as campylobacter jejuni , Listeria monocytogenes were not included in this study. The language should be more brief and to the point.

Please find below other comments throughout the manuscript:

1. safety aspects in the title were not well reflected in the study. The title can be more precise  

2. Section 2.4 and 2.5, simplification the description and provide references instead.

3.  Figure 1a, add % for the Y axis.

4. Please provide some discussion or analysis on the correlation of meat samples and and stool samples.

Author Response

Submission of a revised manuscript biology-1785144

Responses to the reviewer’s comments

Dear Reviewer,

thank you very much for investing you time in this revision and for your comments. Please find the point by point response to your suggestions below.

Reviewer #2:

The manuscript aimed to investigate the microbiological quality and safety of ostrich meat samples. The study provides very useful information for the microbiocidal quality and safety of ostrich meat. However, the data provided in the manuscript was not sufficient, only 50 meat samples were used, and only 3 safety indicators were tested, other important pathogens such as campylobacter jejuni , Listeria monocytogenes were not included in this study. The language should be more brief and to the point.

 Please find below other comments throughout the manuscript:

1. safety aspects in the title were not well reflected in the study. The title can be more precise  

R: The titel was revised accordingly.

2. Section 2.4 and 2.5, simplification the description and provide references instead.

R: Thanks for your comment. We have tried to erase some repetitive information (L121-122, L133-136, L175-180, and L187-190). However, many methods have been changed in-house and can therefore not simply be cited. Methods that are performed according to the manufacturer's instructions have been directly cited.

3.  Figure 1a, add % for the Y axis.

R: The Percent was added, as suggested.

4. Please provide some discussion or analysis on the correlation of meat samples and stool samples.

R: We included one sentence on the differences between stool and meat samples (L395-398).

Reviewer 3 Report

It is a simple and correctly designed manuscript introducing important information on a public health point of view about health status and risks of zoonotic diseases transmission in a species in which farm-level production is increasing as a food supply system in countries shuch as Italy, Germany or Spain.

In order to improve the manuscript, some comments have included in the pdf of the manuscript (see attached document) 

Author Response

Submission of a revised manuscript biology-1785144

Responses to the reviewer’s comments

Dear Reviewer,

thank you very much for investing your time in this revision and for your comments. Please find the point by point response to your suggestions below.

Reviewer #3:

It is a simple and correctly designed manuscript introducing important information on a public health point of view about health status and risks of zoonotic diseases transmission in a species in which farm-level production is increasing as a food supply system in countries shuch as Italy, Germany or Spain.

In order to improve the manuscript, some comments have included in the pdf of the manuscript (see attached document) 

I think it could be very interesting to include specific information about Campylobacter spp because in Europe it is the most prevalent foodoborne disease related to birds in a domestic production level. Did you get any specific data about it?

R: Thanks for your comment. We also consider Campylobacter to be very important, but due to different reasons (financial and collaborative), we did not include more pathogens in the analysis.

Is this represenntative of the animals in the selected fourf farms. The question continues wit the next comment. Criteria for sample size and methods? Is it representative of the ostrich farms in Germany or bavaria? I think important to coment it in order to use the results as a model of the osrich farms in? I think it is important to explain the representativiness of the study

R: Thanks for your comment. We mention in the introduction, that the ostrich population in Germany is of 2,500 animals. We are aware of the fact, that sampling of 50 animals is not determinant for the whole population. It is however extremely difficult in some cases, to acquire enough samples from these animals, as they are still not widely marketed and slaughter is frequently not consistently planned in advance. Therefore, we state throughout the manuscript, that further studies are needed and that the data from this study should be use together with further studies, not only from Bavaria but from the rest of Germany and other countries in Europe. We also consider, that in case of a monitoring program being implemented and establishment of microbiological parameters for ostrich meat takes place, this should be meant for the whole European Union at least. Therefore, information from the main countries involved in ostrich farming would be needed.

I think it could be interesting to get specific data for Campylobacter because its importance in public health. Any possibility to include these data???? The rest of the methodology seems perfect to me. Results are clearly explained and supported in data (absolute and relative) in the case of the studied microorganisms, but it surpissed to em that there are nome reference to Campylobacter.

R: Dear reviewer, thanks for this suggestion. We will surely consider this option, if access to ostrich meat may be possible again. Sadly, amount or samples available and access to the farms was extremely limited and we the total amount of meet and stool were necessary for the analyses. Therefore, it is sadly impossible to analyzed them for this zoonotic pathogen anymore. We will however keep it in mind, should the access to the farms be possible again.